# Relationship between Cholesterol-Related Lipids and Severe Acute Pancreatitis: From Bench to Bedside

**DOI:** 10.3390/jcm12051729

**Published:** 2023-02-21

**Authors:** Xiaoying Zhou, Shengchun Jin, Jingyi Pan, Qingyi Lin, Shaopeng Yang, Yajing Lu, Minhao Qiu, Peter C. Ambe, Zarrin Basharat, Vincent Zimmer, Wei Wang, Wandong Hong

**Affiliations:** 1Department of Gastroenterology and Hepatology, The First Affiliated Hospital of Wenzhou Medical University, Wenzhou 325000, China; 2School of the First Clinical Medical Sciences, Wenzhou Medical University, Wenzhou 325035, China; 3Department of General Surgery, Visceral Surgery and Coloproctology, Vinzenz-Pallotti-Hospital Bensberg, Vinzenz-Pallotti-Str. 20–24, 51429 Bensberg, Germany; 4Jamil-ur-Rahman Center for Genome Research, Dr. Panjwani Centre for Molecular Medicine and Drug Research, International Center for Chemical and Biological Sciences, University of Karachi, Karachi 75270, Pakistan; 5Department of Medicine, Marienhausklinik St. Josef Kohlhof, 66539 Neunkirchen, Germany; 6Department of Medicine II, Saarland University Medical Center, Saarland University, 66421 Homburg, Germany; 7School of Mental Health, Wenzhou Medical University, Wenzhou 325035, China

**Keywords:** cholesterol, acute pancreatitis, HDL-C, SAP, hypercholestrolemia

## Abstract

**Highlights:**

**What are the main finding?**
Higher serum levels of total cholesterol and low-density lipoprotein cholesterol are associated with the severity of AP, while the persistent inflammation of AP is linked with decreased serum levels of cholesterol-related lipids.

**What is the implication of the main finding?**
Cholesterol-related lipids should be recommended both as risk factors and early predictors for studying the severity of AP.Cholesterol-lowering drugs may play a role in the treatment and prevention of AP with hypercholesterolemia.

**Abstract:**

It is well known that hypercholesterolemia in the body has pro-inflammatory effects through the formation of inflammasomes and augmentation of TLR (Toll-like receptor) signaling, which gives rise to cardiovascular disease and neurodegenerative diseases. However, the interaction between cholesterol-related lipids and acute pancreatitis (AP) has not yet been summarized before. This hinders the consensus on the existence and clinical importance of cholesterol-associated AP. This review focuses on the possible interaction between AP and cholesterol-related lipids, which include total cholesterol, high-density lipoprotein cholesterol (HDL-C), low-density lipoprotein cholesterol (LDL-C) and apolipoprotein (Apo) A1, from the bench to the bedside. With a higher serum level of total cholesterol, LDL-C is associated with the severity of AP, while the persistent inflammation of AP is allied with a decrease in serum levels of cholesterol-related lipids. Therefore, an interaction between cholesterol-related lipids and AP is postulated. Cholesterol-related lipids should be recommended as risk factors and early predictors for measuring the severity of AP. Cholesterol-lowering drugs may play a role in the treatment and prevention of AP with hypercholesterolemia.

## 1. Introduction

Cholesterol was first isolated from human gallstones by Poulletier Delasalle in 1769, and its unique physiological and pathological effects have been extensively investigated for more than two centuries [1]. Cholesterol is a hydrophobic sterol molecule determined by three domains structure, involving hydrophilic, hydrophobic and rigid domains [2]. Cholesterol is present in almost every nucleated animal and human cell, and the basic structure accounts for its central role in maintaining rigidity and affecting the cell’s permeability [3].

The basic biological role of cholesterol can be considered from three aspects: cell signaling, hormone synthesis and metabolic homeostasis [4]. At the cellular level, when the membrane cholesterol concentration reaches 10%, it provides organization to the many cellular receptors and transporters, thereby forming dynamic microdomains (which are known as “*rafts*”) that function in endocytosis, membrane signaling and trafficking [4]. In terms of endocrine regulation, cholesterol is converted to biologically active steroid hormones (such as glucocorticoids and mineralocorticoids) in the process of steroidogenesis [5]. Bile acids are the end products of human cholesterol catabolism in the liver, which serve as metabolic integrators to regulate and maintain lipid as well as glucose metabolic homeostasis [6].

In contrast to other metabolites, cholesterol fails to support the production of ATP (adenosine triphosphate). On the other hand, excessive cholesterol might be deleterious or even lethal for nucleated cells [7]. Therefore, the regulation of cholesterol content is tightly monitored by complex feedback mechanisms within physiological ranges (which will be introduced in detail later). The accumulation of cholesterol promotes the activation of both innate and adaptive inflammatory responses, including the augmentation of Toll-like receptor 4 (TLR4) signaling and NLRP 3 (NOD-like receptor thermal protein domain-associated protein 3) inflammasome activation [8].

Acute pancreatitis (AP) is an inflammatory disease that is triggered by disturbances in the pancreatic acinar cell control of pancreatic digestive enzymes in the early stages and followed by multiple parallel mechanisms. This drives a profound systemic inflammatory response and extensive pancreatic auto-inflammation, compelled by the activation of inflammatory cascades and pathologic calcium signaling [9]. TLR 4 acts as its major mediator in the development of pancreatic injury during severe acute pancreatitis (SAP) and has been identified as a potentially promising therapeutic target in AP [10,11]. It has been reported that NLRP3 inflammasome activation is involved in the pathogenesis of AP, and the inhibition of NLRP3 inflammasomes significantly reduces pancreatic organ injury along with systemic inflammation in animals [12]. Given the common involvement of TLR4 signaling and NLRP3 inflammasomes in both hypercholesterolemia and the pathogenesis of AP, recent studies indicated that both high total cholesterol (TC, >240 mg/dL) and high low-density lipoprotein cholesterol (LDL-C, >150 mg/dL) levels within 24 h of admission are independently associated with an increased risk of SAP [13,14]. On the other hand, the development of SAP is associated with a decrease in cholesterol-related lipids, including total cholesterol, high-density lipoprotein cholesterol (HDL-C), LDL-C and Apo (apolipoprotein) A1 [13,14,15]. Therefore, cholesterol-related lipids have been proposed as potential risk factors or predictors for the development of SAP [13,14,15].

To the best of our knowledge, no literature review on the relationship between cholesterol-related lipids and SAP has been published in the literature to date. In this review, we summarize existing clinical and experimental studies to provide the readers an overview of the mechanism and clinical results of the relationship between cholesterol-related lipids and the severity of AP (Figure 1).

## 2. Overview of Homeostasis of Cholesterol-Related Body Lipids

There are two main forms of cholesterol in the human body—free cholesterol (also called unesterified cholesterol) and neutral cholesterol esters (excess cholesterol is esterified by ACAT (acyl-coenzyme A–cholesterol acyltransferase)) [3]. Owing to the special hydrophobicity of cholesterol, it is always found in membranes and stored as a reservoir in cytosolic lipid droplets or combined with Apo to release and metabolize in the form of plasma lipoproteins in the blood [1]. Lipoproteins are further classified on the basis of their particle size, electrophoretic mobility or affinity chromatography. These include HDL-C (d = 1.063–1.210 g/mL), LDL-C (d = 1.019–1.063 g/mL), IDL-C (intermediate density lipoprotein cholesterol, d = 1.006–1.019 g/mL), VLDL-C (very low-density lipoproteins cholesterol, d < 0.006 g/mL) and chylomicrons (d < 0.95 g/mL) [16].

Due to the involvement of different forms of cholesterols, the homeostasis of cholesterol is strictly regulated to ensure proper the functioning of cells, including during absorption, de novo synthesis, excretion, conversion and transport (Figure 2) [1,17,18].

Cholesterol is synthesized via a multi-step enzymatic process called de novo synthesis, in which the HMGR and SM are the rate-limiting enzymes. In addition, cells can absorb cholesterol though the LDLR-dependent endocytosis. Enterocytes and hepatocytes also absorb the cholesterol through the apical NPC1L1 protein. In contrast, cells can excrete cholesterol through ABCA1 and ABCG1, while enterocytes and hepatocytes excrete cholesterol through the apical ABCG5 and ABCG8. Meanwhile, cholesterol can be converted into CE by ACAT, which can be stored in cytosolic lipid droplets or released as plasma lipoproteins. Two transcription factors, sterol regulatory SREBP2 and LXR, play an important role in the regulation of cholesterol homeostasis. SREB2 can be activated when the cellular level of cholesterol is low and can subsequently upregulate the expression of NPC1L1, LDLR, HMGR and SM. LXR can be activated when the cellular level of cholesterol is high and can subsequently upregulate the expression of ABCG5/G8 and ABCA1/G1 and downregulate the expression of NPC1L1. Abbreviations: ABC, ATP-binding cassette transporter; ACAT, acyl coenzyme A; cholesterol acyltransferase; CE, cholesteryl ester; CoA, coenzyme A; HMG-CoA, 3-hydroxy-3-methylglutaryl-CoA; ER, endoplasmic reticulum; HDL, high-density lipoproteins; HMGR, HMG-CoA reductase; LDL, low-density lipoproteins; LDLR, LDL receptor; LXRs, liver X receptors; PP, pyrophosphate; SREBP2, sterol regulatory element-binding protein-2; SM, squalene monooxygenase; NPC1L1, Niemann–Pick C1-like 1.

The absorption from the daily diet and bile is the main source of the cholesterol pool in human body [1]. It is mediated by a transporter named the NPC1L1 (Niemann–Pick C1-like 1) protein, which is expressed in enterocytes and hepatocytes [19]. In contrast, the ABC (ATP-binding cassette) transporters ABCG5 and ABCG8, also expressed in enterocytes and hepatocytes, represent export pumps mediating the excretion of cholesterol and plant sterols [19]. De novo synthesis is also an important endogenous supply for the cholesterol pool in the human body. From HMG-CoA (acetyl-CoA to 3-hydroxy-3-methylglutaryl-CoA) to cholesterol, it is an enzymatic process with a series of steps that occur mainly in the ER (endoplasmic reticulum), within almost all human cells [20].

Cholesterol absorbed by enterocytes is subsequently converted to chylomicrons, which get released into the circulatory system and finally transported to the liver [1]. Cholesterol biosynthesis also occurs in the liver, where about 50% of the total newly generated cholesterol is produced [21]. As the center of cholesterol homeostasis, exogenous and endogenous cholesterol are transported into the bloodstream in the form of VLDL-C [1]. Releasing the triglyceride content by degrees, VLDL-C turns into LDL-Cs, which can be taken up by cells via LDL-C receptor-dependent endocytosis [18].

To prevent general cell toxicity, excess cholesterol is disposed of in two ways: first via esterification by ACAT to cholesteryl esters, which can be stored in cytosolic lipid droplets or released as plasma lipoproteins, and second via transport outside cells through the ABCA1/ABCG1 and conversion into HDL-C [1,2,3]. Most HDL-Cs are eventually transported from peripheral tissues back to the liver, where cholesterol can be recycled or excreted, which is called reverse cholesterol transport (RCT) [1].

The regulation of cholesterol homeostasis is tightly governed through complex transcriptional and post-translational interplays. The two main transcription factors, SREBP2 (sterol regulatory element–binding protein-2) and LXRs (liver X receptors), are vital in this regulatory process [3,22].

When cellular cholesterol levels are low, SREBP2 can be proteolytically processed into an active transcription factor in the Golgi, which subsequently enters the nucleus and promotes the transcription of genes of key elements in cholesterol homeostasis [3,18,23]. These key elements include two rate-limiting enzymes of the de novo biosynthesis, HMGR (HMG-CoA reductase) and squalene monooxygenase, along with the LDL-C endocytosis mediator LDL-C receptor and the cholesterol import molecule NPC1L1 [1].

The nuclear receptors LXRs can be activated when cellular cholesterol levels are high and subsequently upregulate the expression of the cholesterol export molecules ABCA1, ABCG1, ABCG5 and ABCG8, as well as downregulating the expression of import NPC1L1 molecules [18,24].

## 3. Mechanism of Interaction between Cholesterol-Related Lipids, Inflammation and AP

### 3.1. Hypercholesterolemia, LDL-C May Aggravate the Inflammation and Severity of SAP

Hypercholesterolemia is an enrichment of cholesterol in plasma, and a serum total cholesterol level of more than 240 mg/dL is one of the defining indicators of hypercholesterolemia [25]. In macrophages and other immune cells, cholesterol accumulation promotes inflammatory reactions by enhancing the TLR signal, enabling inflammasomes and producing monocytes and neutrophils in the bone marrow and spleen [8].

In macrophages, the center of the innate immune response, the downstream mechanism of inflammation has been studied mainly under TLR4 stimulation [3]. Cholesterol enrichment in the plasma membrane promotes the formation of receptor complexes that form and enhance LPS (lipopolysaccharide)-mediated signaling by providing focused signaling events, such as the LPS response to TLR4 [26]. In addition to LPS stimulation, including TLR4, HDL-C-associated proinflammatory responses have been found to strengthen TLR7/8 in vivo [27]. The LXRs could eliminate excess cholesterol by providing a feedforward regulation system [7], while existing studies suggest that the expression of LXR target genes can be inhibited through the activation of TLR [28]. Moreover, the activation of TLR also inhibits the ability of macrophages to expel excessive cholesterol [28], thereby amplifying TLR signaling [8]. The activation of TLR reduces cholesterol efflux, aggravates cholesterol accumulation in macrophages and amplifies the inflammatory response [8].

Pancreatic microvascular abnormalities and ischemia also play an important role in the early pathogenesis of AP [29]. The histopathological results reveal microcirculatory intravascular thrombosis, intravascular stasis and endothelial desquamation, as well as parenchymal swelling of the pancreas allied with the development of pancreatic necrosis [29]. Hypercholesterolemia may damage vascular endothelial cells to synthesize nitric oxide and cause endothelial progenitor cell dysfunction in addition to endothelial microinflammation [30,31], which ultimately reduces the blood supply and aggravates the severity of the disease. It has been reported that oxidized LDL correlates with thrombogenesis by interfering with the coagulation system and clot formation, which may deteriorate organ ischemia [32].

Cholesterol crystallization is one of the known stimuli of inflammasome activation and may also be one of the second signals of inflammasome activation in macrophages [33,34]. Both intracellular and extracellular cholesterol can activate NLRP (NOD-like receptor thermal protein domain-associated protein)-3 inflammasomes and increase their initiation [35,36]. This leads to activation of the caspase-1- and secretion of IL (interleukin)-1 family cytokines [37,38]. IL-1 cytokines share intracellular signaling domains with TLRs, inducing a highly pro-inflammatory genetic program downstream of the adaptor molecule (major reactive protein 88) to produce additional inflammatory mediators that exacerbate the inflammatory response [39].

Westerterp et al. reported that cholesterol deposits promote neutrophil accumulation and the formation of NETs (neutrophil extracellular traps) after the activation of NLRP3 inflammasomes in myeloid cells [34]. It has been observed that neutrophils treated with different concentrations of methyl-β-cyclodextrin to reduce cellular cholesterol levels have shown significant release of NETs [40]. Cholesterol crystals could promote the formation of NETs together with IL-1-β [3]. Cholesterol crystals induce neutrophils to release NETs, which in turn stimulate macrophages to release cytokines that activate TH17 (T-helper 17) cells [41]. In addition, the interaction of danger signals with neutrophils may subsequently lead to sterile inflammation [41]. The mechanism of cholesterol leading to inflammation through TLRs, NLRP3 and NETs is shown in Figure 3. The relationship between TLRs, NLRP3 and NETs in the proinflammatory process is also shown in detail.

Several pathways of cholesterol-mediated inflammation have been described above, with AP also being a form of inflammation. The pathogenesis of AP has been extensively studied. At the cellular level, the critical events include pathological calcium signaling, the premature activation of trypsinogen, ER stress, mitochondrial dysfunction, impaired UPR (unfolded protein responses) and impaired autophagy [42,43,44]. Additionally, TLR4, NLRP3 and NETs are also activated by cholesterol to induce inflammation and play an important role in the pathogenesis of AP [45].

The following paragraph will briefly describe the mechanisms mentioned above. Typical damage-associated molecular patterns released from injured pancreatic cells, such as high mobility group box-1 and heat shock protein 70, stimulate NF-κB (nuclear factor kappa-B) activation via TLR4, further upregulating the expression of NLRP3, pro-IL1β and pro-IL18 [24]. Hoque et al. have shown that components of NLRP3 inflammatory are required for severe pancreatic injury [46]. Precursors of IL1β and IL18 cytokines are converted into active forms via NLRP3 inflammasomes. NETs can be involved in AP by triggering the activation of trypsin through STAT-3, MMP-9 or other pathways, promoting local and distal organ damage and being a central part of AP progression [47].

In the pathogenesis of AP, calcium, ER stress and UPR are not independent of each other. In fact, they are inextricably linked to cholesterol. Feng et al. have illustrated a signal transduction pathway that connects free cholesterol enrichment with the triggering of apoptosis [48]. Due to the particularly low cholesterol content of the ER membrane, the function of this organelle is likely to be exceptionally sensitive to abnormal accumulation with free cholesterol [49]. The ER, not the plasma membrane, is most consequential in the process of cholesterol-induced apoptosis [48]. In addition to processing, transporting and storing proteins, the ER is also an important organelle for intracellular calcium storage [50]. Abnormal loading of cholesterol in the ER membrane leads to the exhaustion of ER calcium stores and ER stress, initiating the UPR and ultimately apoptosis [48].

The UPR is a specific signaling pathway from the ER, a cytoprotective mechanism that responds to ER stress by altering transcriptional and translational programs [51]. The cholesterol load of the ER-membrane-mediated cell death depends on the release of the ER-chaperone-binding immunoglobulin protein, which activates the UPR sensors protein kinase RNA-like ER kinase and inositol-requiring enzyme 1 [52]. Activated protein kinase RNA-like ER kinase mediates the translation of transcription factor (activating transcription factor 4) and promotes the expression of pro-apoptotic factor CHOP (CEBP/EBP homologous protein) [53]. Additionally, activated inositol-requiring enzyme 1 induces a mitochondria-dependent caspase cascade. In addition, free cholesterol can trigger apoptosis though the Fas pathway [53] (Figure 4).

Autophagy is a cytoprotective mechanism, which plays an important role in maintaining cellular integrity by eliminating defective or damaged cell contents and misfolded proteins from cells. However, the disordered autophagy pathways induce pancreatitis [54,55,56]. The mannose-6-phosphate pathway plays an important role in the lysosomal/autophagic pathway and pancreatic cholesterol metabolism [57]. Defects in the mannose-6-phosphate pathway disrupt cholesterol conversion in islet cells, allowing non-esterified cholesterol to accumulate in lysosomes/autolysosomes and be consumed in the plasma membrane, leading to increased cholesterol synthesis and uptake. In addition, its defect leads to a significant dysfunction of the mitochondria in the pancreas and a significant increase in pancreatic protease activity, both of which are mechanisms for the development of AP [57].

The evidence shows that a cycle of intracellular cholesterol hydrolysis and re-esterification continues in a natural way [58]. However, when cholesterol accumulates in excess for various reasons, this cycle is disrupted and free cholesterol levels in the ER and mitochondria increase. This leads to events such as mitochondrial dysfunction, oxidative stress, the sensitization of cells to cytokines and cell death, as mentioned above [59,60]. Taken together, this probably explains why high cholesterol can exacerbate AP.

The mechanisms of high LDL-C to aggravate SAP may include increasing oxidative stress and amplifying the inflammatory response. LDL-C may increase ROS (reactive oxygen species) and reduce nitric oxide [61]. Nitric oxide has anti-inflammatory effects. It can inhibit the nuclear transcription factor NF-κB, a key regulator of cytokine induction and production, reduce cytokine release and reduce mitochondrial ROS production by limiting mitochondrial oxidative phosphorylation [61]. ROS can not only induce tissue damage, but can also activate NF-κB together with lipid peroxidation and contribute to the production of cytokines and chemokines, such as IL-6 and monocyte chemokine 1 [62]^,^ [63]. These two effects of ROS are ultimately involved in the mechanism of mild AP developing into SAP. On the other hand, some studies have shown that minimally modified LDL-C can bind to TLR2 and TLR4 and induce the release of proinflammatory factors [64]. Meanwhile, a large amount of freely oxidized LDL-C in the circulation can combine with specific antibodies to form immune complexes and induce inflammatory responses in macrophages and dendritic cells [65].

### 3.2. Persistent Inflammation and AP May Reduce the Serum Cholesterol-Related Lipids

While cholesterol contributes to the development of inflammation, inflammation could also have an impact on cholesterol levels. Inflammation causes acute phase responses that lead to changes in lipid metabolism. Amongst the primates, as represented by humans, infection or inflammation has been found to reduce serum cholesterol levels [66].

It is worth noting that clinical studies have also shown that patients with SAP have lower total cholesterol concentrations (3 to 15 days after admission) compared to non-SAP patients [14]. There are several possible explanations for this: first, because of fasting in patients with AP, with reduced external nutritional intake and increased catabolism, leading to lower serum cholesterol concentrations [67]; second, because of the extremely high similarity, an analogy can be drawn with severe sepsis [68]. It was hypothesized after studying patients with severe sepsis that reduced hepatic cholesterol synthesis is strongly associated with the excessive release of inflammatory cytokines early in AP [69]. Finally, active inflammatory cytokines such as IL-6 and TNF-α (tumor necrosis factor-α) also damage the microvascular endothelium, leading to increased capillary permeability and the spillage of lipoproteins from the vessels into the interstitium, which may also be a mechanism for lower total cholesterol levels [14,70].

The results from standard clinical trials and studies related to inflammation have demonstrated reductions in total cholesterol and LDL-C cholesterol [71]. During inflammation, increased LDL-C receptor expression leads to decreased LDL-C levels [38]. The LDL-C receptor mediates the uptake of plasma LDL-C by cells and the degradation of plasma LDL-C. It is one of the important determinants of plasma cholesterol levels. Meanwhile, studies have demonstrated the stimulatory effects of IL-1β, IL-6 and TNF-α on LDL-C receptor activity [7]. Lubrano et al. found that LDL-C is mostly converted to ox-LDL-C in hypercholesterolemic conditions and that increased ox-LDL-C in the presence of inflammation promotes oxidized lipoprotein receptor-1 expression. This may explain the rapid reduction in cholesterol in the acute phase of inflammatory diseases, and it is also noteworthy that oxidized lipoprotein receptor-1 is particularly responsible for the reduction in LDL-C levels [52]. In the process of the inflammatory response, C-reactive protein, which is a marker of the inflammatory response, can combine with LDL-C cholesterol to form a complex that can be quickly broken down, thereby reducing the level of LDL-C [72] (Figure 5). Lower LDL-C levels are also correlated with a higher severity of AP [58]. The univariate logistic regression analysis by Chen et al. also illustrated that the risk of pancreatic necrosis was related to low levels of LDL-C [63]. One possible explanation for this phenomenon is that the excessive emission of inflammatory cytokines, such as IL-6 and TNF-α, early in AP may lead to lower hepatic LDL-C synthesis [73]. Another possible explanation is that increased capillary permeability in AP leads to the redistribution of lipoproteins from intravascular to extravascular environments [69].

The serum concentration of HDL-C is significantly reduced in the inflammatory state, and the structural and functional changes of HDL-C may play an important regulatory role [74]. The specific mechanism by which inflammation reduces serum HDL-C and Apo A1 levels is uncertain, but there are many possible pathways involved [75]. It has been revealed that IL-1, IL-6 and other inflammatory stimulators may reduce the expression of Apo A1 in the acute phase reaction [76]. It should be noted that the decrease in circulating Apo A1 may lead to a drop in circulating HDL-C cholesterol [74]. SAA (serum amyloid A) is an acute phase reactant that is significantly increased in the acute response phase [77]. During the acute phase response, SAA significantly increases while Apo A1 decreases [78]. SAA replaces Apo A1 in HDL-C particles as the main apo-lipoprotein and accelerates the clearance of HDL-C in the circulation (Figure 4) [74,78]. Another important mechanism is that cytokines promote endothelial lipase activity under inflammatory conditions, thereby contributing to the reduction in HDL-C [79].

There has been considerable evidence indicating that RCT is also impaired during the acute phase responses [79]. ABCA1 mediates the transfer of cholesterol from macrophages to HDL-C, and inflammatory stimulation reduces the expression of ABCA1 mRNA during the acute phase response, which inhibits cholesterol efflux [80]. Inflammation also inhibits ABCG5 and ABCG8 mRNA and protein levels during hepatic RCT, which weakens cholesterol excretion from the liver to bile (Figure 4) [81]. HDL-C mediates cholesterol clearance from cells and has an initial role in RCT [15]. In addition, the previously mentioned SAA replaces Apo A1 as the main apolipoprotein in HDL-C, which has an impaired ability to regulate cholesterol efflux [82]. Khovidhunkit et al. incubated murine macrophages with acute-phase HDL-C and detected the cholesterol content, and the experimental results showed that the cholesterol efflux decreased and intracellular cholesterol content increased, which also indicated that the ability of HDL-C to clear cholesterol was impaired in the acute phase [15]. During inflammation, TLR activation also inhibits the ability of macrophages to expel excess cholesterol [28].

Apo A1 is the main structural protein of HDL-C and plays an important role in RCT and the activation of cholesterol esterification enzymes [83,84]. It is also involved in the regulation of inflammatory and immune response, while its anti-inflammatory role could be a protective factor for AP [85,86]. Navarro et al. conducted an experiment in which turpentine oil was injected into pigs to induce an acute inflammatory response, and the results showed a gradual decrease in plasma Apo A-I levels, along with decreased mRNA expression of Apo A-I [76]. Using Apo A-I-deficient mice but with no significant difference in HDL-C levels between different genotypes as models to experiment, the results after the injection of LPS proved that the LA (LDL-CR^-/-^/Apo bec^-/-^)-Apo A-1^-/-^ mice had increased sensitivity to inflammation due to their greater production of proinflammatory cytokines and chemokines [87]. Overexpressed inflammatory cytokines such as TNF-α and IL-6 can decrease the production of Apo A1 [88,89].

## 4. Clinical Results of Relationships between Cholesterol-Related Lipids and AP

As mentioned above, we intend to discuss the research on total cholesterol and lipoprotein with AP. However, the research concentrating on the potential role of VLDL-C and chylomicron is still sporadic, limiting our discussion in this specific aspect. Meanwhile, Apo A1 serves as the major component of HDL-C, and its potential importance as an anti-inflammatory agent has been demonstrated in recent studies.

### 4.1. Clinical Results on the Relationship between Total Cholesterol and SAP

Considering that total cholesterol could induce inflammation by enhancing TLR signals and enabling inflammasomes, the high level of total cholesterol presents an inducive effect at the early stage of AP onset. As expected, a prospective analysis in a multicentric cohort study affirmed this view by observing an increased risk of gallstone-induced AP with higher cholesterol intake levels [90]. The increasing level of cholesterol is associated with the development of SAP and a poor clinical outcome [65,91,92]. Shen et al. conducted a case–control study and suggested that the occurrence of AP was significantly associated with high cholesterol (≥220 mg/d, OR 1.992) [93]. One retrospective research enrolled 648 patients with AP and found that cholesterol was significantly associated with the development of SAP [14]. In addition, it was reported that maternal and fetal morbidity are highly associated with the severity of AP and cholesterol. This was incorporated into a nomogram to predict moderately severe and severe AP in pregnancy (ROC of 0.865 and 0.853 in the training and validation sets, respectively) [90]. It achieved a sensitivity rate of 0.868 and specificity rate of 0.771 in the training sets, alongside a sensitivity rate of 0.812 and specificity rate of 0.875 in the validation sets [90]. Similarly, a ten-year retrospective cohort experience among women with AP in pregnancy indicated that SAP patients had significantly higher total cholesterol levels when compared to mild AP and mild SAP [92]. In their study, they also suggested that total cholesterol could be a significant risk factor influencing the length of hospital stay [92]. Düzenci et al. analyzed the records of 153 AP patients and found a significant correlation between cholesterol and mortality [94]. Meanwhile, another retrospective research study conducted by Song et al. reported that the cholesterol level was significantly higher in recurrent AP compared to AP at first onset [95].

On the contrary, an observational study reported that the total cholesterol measured within 2 days of admission was significantly lower in alcohol-induced AP patients compared to alcoholic patients. It was also associated with in-hospital mortality and longer hospitalization stay [96]. In a study by Peng et al., based on the relatively later data collection for the total cholesterol (with a median of 3 days (IQR 2 to 4 days) after the start of symptoms and a median of 2 days (IQR 1 to 3 days) after hospital admission), the total cholesterol level was significantly lower in AP patients with organ failure than without organ failure [97].

Hong et al. suggested that there was a U-shaped association between total cholesterol levels assessed within 24 h of admission and SAP using a restricted cubic spline analysis, which indicates that either low total cholesterol levels (<160 mg/dL, OR 2.72) or high total cholesterol levels (>240 mg/dL, OR 2.54) within 24 h of admission had a higher risk of SAP [14]. In addition, a longitudinal cohort follow-up of these patients suggested that SAP patients had a reduced level of total cholesterol within 3–15 days during hospitalization compared to non-SAP controls [14]. Through early blood samples assayed within 24 h and a longitudinal cohort follow-up, this study demonstrated that patients with hypercholesterolemia had a high risk of developing SAP, while the persistent inflammation of AP could be due to a decrease in the total cholesterol concentration.

The existing clinical research on total cholesterol and AP is summarized in Table 1.

### 4.2. Clinical Results of Relationships between LDL-C and SAP

After analyzing the records of 157 hypertriglyceridemic pancreatitis patients in the study by Liao et al., higher LDL-C was found to be associated with the recurrence of hypertriglyceridemia-induced AP [99]. In addition, the serum amylase level of AP patients after 5–7 days of onset or in patients with alcohol-induced AP, appeared to be less than three times the upper limit of normal level and led to SAP [100]. It was reported that a higher LDL-C level was independently associated with the development of AP along with low serum amylase levels (OR 1.009) [100].

Many studies have shown a decrease in serum LDL-C levels following the onset of AP. Khan et al. reported lower serum LDL-C levels within 2 days after admission in SAP patients, which was associated with a higher rate of in-hospital mortality and longer hospital stay (r = −0.264) [96]. In a recent study conducted by Hong et al., LDL-C was found to be associated with the development of SAP according to the 2012 revised Atlanta classification [91]. Peng et al. found that the serum LDL-C level of patients with persistent organ failure was significantly lower than that of patients without persistent organ failure within 24 h of admission, where persistent organ failure was closely related to the high mortality rate of SAP [97]. Wu et al. conducted a retrospective database cohort study with 166 AP patients admitted to the ICU and reported a significant decrease in serum LDL-C levels in non-survivors compared to survivors [101].

Recently, Hong et al. reported a non-linearity relationship between LDL-C and the incidence of SAP using a restricted cubic spline analysis [13]. They suggested that both low LDL-C (<90 mg/dL; OR 3.05) and high LDL-C (>150 mg/dL; OR 4.42) levels within 24 h from admission were independently associated with an increased risk of SAP [13]. These results suggested that LDL-C patients had a high risk of developing SAP, while the persistent inflammation of AP could arise with a decrease in LDL-C.

The currently available clinical research on LDL-C and AP is summarized in Table 2.

### 4.3. Clinical Results of Relationships between HDL-C, Apo A1 and SAP

Several studies have shown that the low levels of the HDL-C and its main component Apo A1 at admission were negatively associated with the severity of AP, which mostly was defined according to the 2012 revised Atlanta classification and poor clinical outcomes [15,62,83,96,97,102]. Additionally, HDL-C and Apo A1 could be biomarkers for predicting persistent organ failure [15,97].

Khan et al. analyzed the records of 233 patients hospitalized for AP and found that the HDL-C levels were significantly lower in patients with SAP, which was defined according to the Atlanta Symposium in 1992. Low HDL-C levels were associated with higher in-hospital mortality and longer hospitalization [96,103]. A prospective cross-sectional study in 66 patients with predicted SAP suggests that low levels of HDL-C and Apo A1 within 24 h of admission to the ICU were related to high levels of inflammatory cytokines, persistent organ failure, infected necrosis and hospital mortality [97]. It was also found that HDL-C and Apo A1 could serve as biomarkers for the differentiation of persistent and transient organ failure [97]. As reported earlier, the sensitivity value of HDL-C to predict persistent organ failure among organ failure patients was 0.886 and the specificity value was 0.850, while the sensitivity value of Apo A1 was 0.943 and the specificity value was 0.850 [97]. Zhou et al. analyzed the records of 102 adult AP patients with organ failure, worsening of previous comorbidities or local complications during hospitalization, leading to the conclusion that low HDL-C or Apo A1 levels within 48 h of hospitalization were negatively related to several scoring systems and adverse clinical outcomes [15]. The concentrations of HDL-C and Apo A1 were significantly lower in patients with SAP in contrast to the patients with mild SAP [15]. For the prediction of persist organ failure among the organ failure patients, HDL-C showed a sensitivity rate of 80.77% and a specificity rate of 77.50%, while Apo A1 showed a sensitivity rate of 73.08% and a specificity rate of 100% [15]. After analyzing the records of 647 AP patients, Hong et al. found out via multivariate analysis that HDL-C at admission was independently associated with SAP (OR 0.95) [62]. They also verified that HDL-C achieved a sensitivity rate of 57.1% and a specificity rate of 90.5% to predict the severity of AP, indicating that HDL-C was a useful predictor of SAP [62]. In a cross-sectional study of 1127 patients with AP, Li et al. noted that the baseline serum concentrations of HDL-C and Apo A1 after admission were negatively associated with the incidence of SAP after the adjustment of other covariates [83]. As for the value of predicting SAP, HDL-C showed a sensitivity rate of 0.757 and a specificity rate of 0.551, while Apo A1 showed a sensitivity rate of 0.747 and a specificity rate of 0.641 [83].

Currently available clinical research on HDL-C, Apo A1 and AP is summarized in Table 3.

## 5. Clinical Significance and Future Prospects

Hypertriglyceridemia is the third most common cause of AP and has been widely studied [102]. Triglycerides and cholesterol are both important participants in lipid homeostasis and likely intricately linked to the development or sustainment of inflammation [104]. However, the research focusing on hypercholesterolemia is still sparse. The current review indicated that hypercholesterolemia may be associated with AP [14]. Although a serum total cholesterol level > 240 mg/dL has been identified as a criterion of hypercholesterolemia for individuals according to the guidelines [105], a consensus on the diagnosis criteria for hypercholesterolemia-related AP has not been established. Song et al. reported that cholesterol-induced toxicity in multiple diseases through cholesterol accumulation in the mitochondria causes mitochondrial dysfunction and induces an increase in reactive oxygen species, leading to the activation of the NLRP3 inflammasome and apoptosis promotion by inducing ER stress and mitochondrial dysfunction [18]. However, scarce data are available on the pathogenesis of hypercholesterolemia-related pancreatitis. Therefore, further animal models and cell experiments on this topic would be necessary and interesting.

Given the mechanism of cholesterol-induced inflammation, cholesterol-lowering drugs may play a role in the treatment and prevention of AP with hypercholesterolemia. Kim et al. reported that guggulsterone, a plant steroid, exhibits anti-inflammatory and cholesterol-lowering effects in cerulein-induced AP via the inhibition of extracellular-signal-regulated protein kinase (ERK) and c-Jun N-terminal kinase (JNK) activation [106]. Lipid-lowering therapy with statins as a first-line treatment is a mainstay of hypercholesterolemia treatment [107]. Simvastatin, a HMG-CoA reductase inhibitor, is commonly administered to treat hypercholesterolemia. Matalka suggested that it prevents L-arginine-induced acute toxicity of the pancreas [108]. Simvastatin may decrease the inflammation of the pancreas by modulating the immune response responsible for inflammation [57]. Gornik et al. further suggested that prior statin treatment significantly reduces morbidity and mortality in AP [109]. Lee et al. suggested that statin use is associated with a decreased severity of AP, observed as reductions in both the overall multisystem organ failure incidence and new multisystem organ failure [110]. Statin usage reduces the risk of developing diabetes mellitus after AP [111]. Patients taking statins who suffer from an episode of AP are more likely to follow a mild course and have a lower risk of developing local complications and systemic inflammatory response syndrome [112]. Recently, a randomized phase II trial (NCT02743364) was performed to evaluate whether simvastatin could reduce pancreatitis in patients with recurrent, acute or chronic pancreatitis or not. Therefore, both basic and clinical studies suggest that statins may prevent and even ameliorate AP [113]. The use of drug combinations rather than increasing doses of one drug can achieve greater efficacy and lower risks. A randomized trial suggests that moderate-intensity statin with ezetimibe combination therapy is not inferior to high-intensity statin monotherapy in the treatment of patients with atherosclerotic cardiovascular disease (with LDL-C cholesterol concentrations), while reducing adverse effects [114]. However, the statin with ezetimibe combination therapy has not been evaluated in patients with hypercholesterolemia. In contrast, several studies suggest that statin use seems to be associated with an increased risk of AP [115,116]. Therefore, more prospective randomized controlled trials are needed to determine the efficacy of statin drugs in the treatment of AP with hypercholesterolemia [117]. In addition, future studies are also needed to assess whether lowering the appropriate thresholds (240 mg/dL) of initiating cholesterol-lowering agents will be effective in reducing the risk of AP.

It is well-known that coagulation abnormalities are associated with severity of disease in AP. Pruller et al. reported that trunk-weighted obesity, cholesterol levels and low-grade inflammation are the main determinants for enhanced thrombin generation [118]. Furthermore, de Laat-Kremers et al. suggested that increased BMI and blood lipid levels are associated with a hypercoagulable state, and this hypercoagulability may partly explain the increased risk of cardiovascular disease in individuals with obesity or dyslipidemia [119]. Undas et al. suggested that simvastatin depresses blood clotting by inhibiting the activation of prothrombin, factor V and factor XIII and presents antithrombotic and anti-inflammatory effects in patients with hypercholesterolemia [120]. It was noted that high-dose statin therapy reduces plasma concentrations of soluble P-selectin, a well-established platelet activation marker, in association with an improved fibrin clot phenotype. This highlights the contribution of platelet-derived proteins to a prothrombotic state in hypercholesterolemia and statin-induced antithrombotic effects [121]. A recent clinical study indicated that the use of low-molecular-weight heparin in the early stage of SAP significantly reduces the chance of disease progression and systemic complications [122]. It will be interesting to conduct a clinical study aimed at whether combining low-molecular-weight heparin and statins in the treatment of SAP with hypercholesterolemia shows a synergistic effect to improve prognosis.

Our review indicated that higher serum levels of total cholesterol and LDL-C were associated with an increased severity of AP, while the persistent inflammation of AP was associated with a decreased serum concentration of cholesterol-related lipids. Therefore, cholesterol-related lipids could be used as potential risk factors and early predictors for the severity of AP. Similar to other single predictors, such as admission hematocrit, blood urea nitrogen and cholesterol-related lipid levels, this may lack high sensitivity or specificity [123]. It is hypothesized that combining several clinical parameters may improve the predictive accuracy [123]. Therefore, it will be interesting to integrate cholesterol-related lipids into clinical scoring systems for the prediction of SAP.

## 6. Conclusions

In this review article, we discussed the interactions between cholesterol-related lipids and AP.

A disordered cholesterol metabolism may lead to an enhanced inflammatory response through a strengthened TLR pathway or inflammasome activation. Regarding the cholesterol metabolism and AP, both of which are closely related to inflammation, the disordered cholesterol metabolism could play an important role in the development of AP. On the other hand, AP could also disturb the cholesterol metabolism, leading to impaired RCT and decreased lipid levels. To give a better understanding of the interaction, we reviewed the existing studies to explore the possible mechanism of the relationship between cholesterol-related lipids, inflammation and AP. With a higher serum level of total cholesterol, LDL-C was associated with increased severity of AP, while persistent inflammation of AP was associated with decreased serum levels of cholesterol-related lipids (which include total cholesterol, HDL-C, LDL-C and Apo A1). Meanwhile, the results of the existing clinical studies demonstrate relationships between cholesterol-related lipids and the severity of organ failure or the mortality of AP, which supports the current view of the interaction with cholesterol-related lipids. These should be recommended both as risk factors and early predictors for the severity of AP.

There are some limitations of the current studies about the relationships between cholesterol-related lipids and SAP. First, the exact mechanisms of the interactions between the cholesterol-related lipids and AP are far from being fully understood due to lack of high-quality experimental studies on cell lines, animal models or humans. Additionally, the knowledge on the natural history of cholesterol-related AP is also scarce. However, despite the limited preclinical mechanistic evidence, prevention strategies and treatment targets based on cholesterol pathways have been put forward. Second, cholesterol-related lipids could be used as potential risk or predictive factors for determining the severity of AP. Further prospective evaluation and validation in large independent population sets is still needed. Taken together, the insights into the role of cholesterol-related lipids in the setting of AP are still in their infancy and further research is needed. A better understanding of the interaction between cholesterol-related lipids and AP may have huge potential for prevention, diagnosis and treatment.

## Figures and Tables

**Figure 1 jcm-12-01729-f001:**
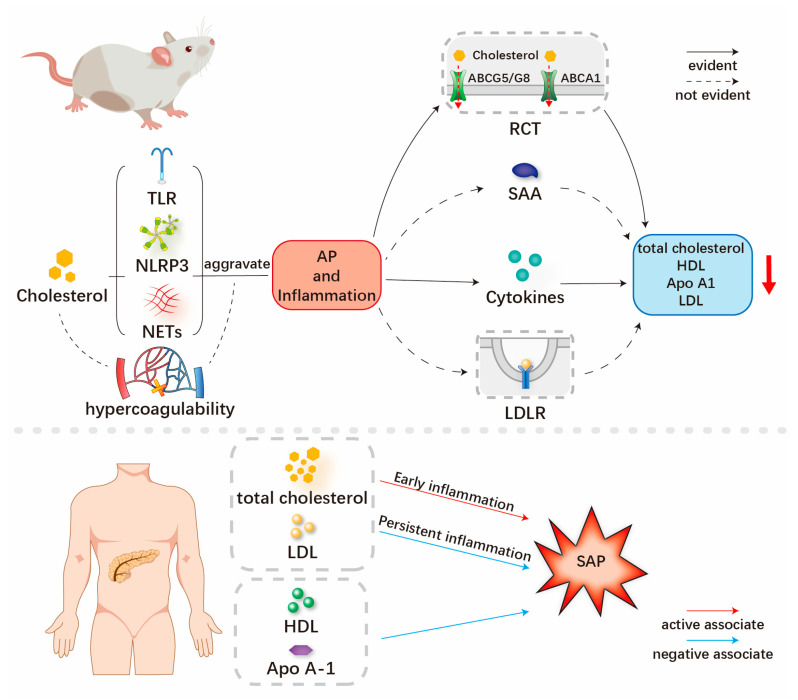
Overview of relationship between cholesterol-related lipids and severe acute pancreatitis.

**Figure 2 jcm-12-01729-f002:**
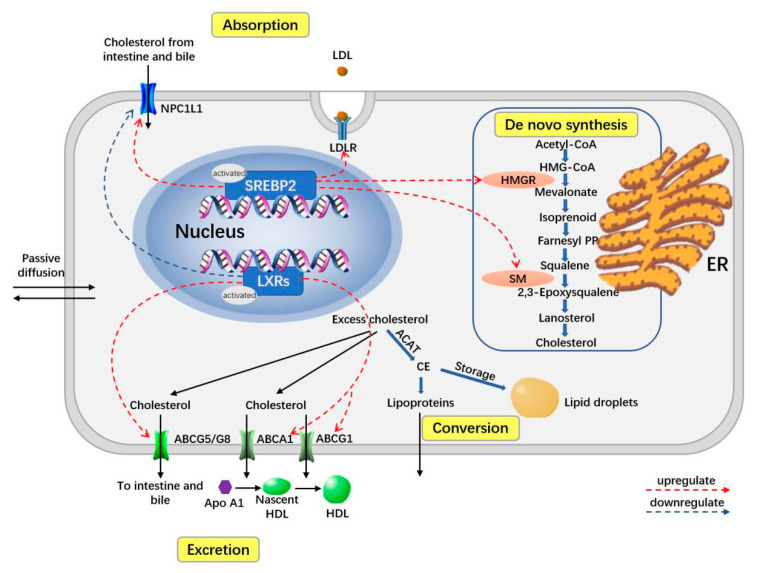
Major pathways of cellular cholesterol homeostasis.

**Figure 3 jcm-12-01729-f003:**
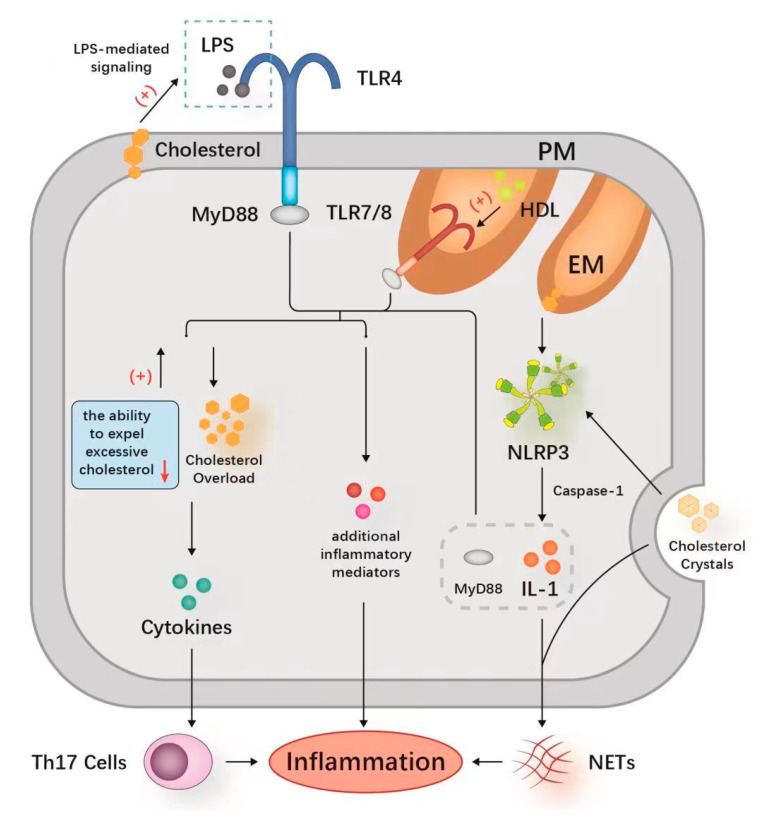
Mechanisms of inflammation caused by abnormal cholesterol. Cholesterol could promote inflammation by activating the TLR signaling pathway to facilitate the release of pro-inflammatory cytokines, and the TLR signaling could be amplified by excess cholesterol in macrophages to exacerbate the inflammatory response. The NLRP3 can be activated by cholesterol and releases IL-1 via the caspase-1-mediated pro-inflammatory IL-1 signaling pathway. IL-1 released after NLRP3 activation shares the intracellular signaling domain with TLRs, inducing MyD88 to produce additional inflammatory mediators that aggravate the inflammatory response. Cholesterol induces the release of NETs, which in turn stimulate macrophages to produce cytokines that activate TH17 cells. The formation of NETs is also associated with the activation of NLRP3. These mechanisms of action do not exist independently but instead are closely related to each other, thereby exacerbating the cholesterol-induced inflammatory response. **Abbreviations**: TLR, toll-like receptor; LPS, lipopolysaccharide; TIR, Toll/IL-1 receptor; IL-1β, interleukin-1β; MyD88, major reactive protein 88; NETs, neutrophil extracellular traps; PM, plasma membrane; ER, endoplasmic reticulum; TH17, T-helper cell 17; HDL, high-density lipoprotein.

**Figure 4 jcm-12-01729-f004:**
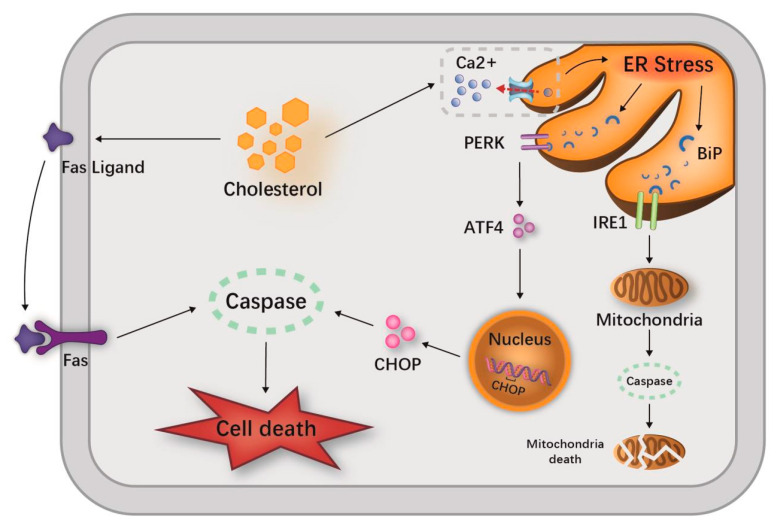
Several pathways of cholesterol-induced cell death. The abnormal loading of cholesterol in the ER membrane leads to the exhaustion of ER calcium stores and ER stress, initiating the UPR and ultimately apoptosis, as well as the release of the ER chaperone BiP, which activates the UPR sensors PERK and IRE1. Activated PERK mediates the translation of ATF4 and promotes the expression of CHOP. Activated IRE1 induces a mitochondria-dependent caspase cascade. ER stress also promotes caspase-12 cleavage and activation, initiating caspase activation and leading to programmed cell death. Free cholesterol can also trigger apoptosis though the Fas pathway. **Abbreviations:** IRE1, inositol requiring enzyme 1; BiP, binding immunoglobulin protein; PERK, protein kinase (PKR)-like ER kinase; ER, endoplasmic reticulum; ATF4, activating transcription factor 4; CHOP, C/EBP homologous protein; ER, endoplasmic reticulum; UPR, unfolded protein responses.

**Figure 5 jcm-12-01729-f005:**
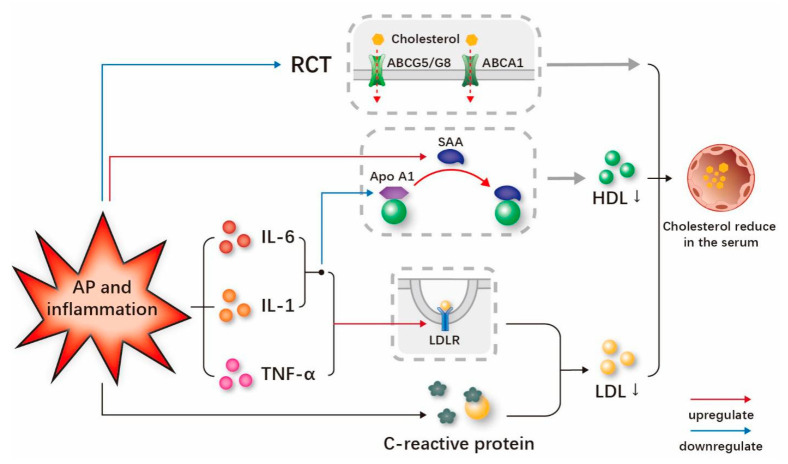
Mechanisms of serum cholesterol reduction caused by persistent inflammation and AP. In the process of persistent inflammation and AP, the release of IL-1, IL-6 and TNF-α increases the expression and activity of LDL-C receptors and leads a decreased LDL-C level. C-reactive protein, a marker of inflammation, also combines with LDL-C to form a complex that can be rapidly degraded to lower LDL-C levels. The excessive release of IL-1 and IL-6 also decreases the serum level of Apo A1. With the decrease in the Apo A1 level, the SAA level increases, and SAA replaces Apo A1 as the main lipid in HDL-C. This change in HDL-C speeds up the clearance of HDL-C from the circulation and leads to a decrease in HDL-C levels. RCT is also damaged in the process of inflammation and AP, and the decreased expression of ABCA1, ABCG5 and ABCG8 mRNA blocks the efflux of cholesterol, which is also one of the mechanisms of the decrease in the serum cholesterol level. **Abbreviations**: IL-6, interleukin-6; IL-1, interleukin-1; TNF-α, tumor necrosis factor-α; HDL-C, high-density lipoprotein cholesterol; LDL, low-density lipoprotein cholesterol; SAA, serum amyloid.

**Table 1 jcm-12-01729-t001:** Characteristics of included clinical studies related to the relationship between total cholesterol and SAP.

Author and Year of Publication	Type of Study	Sample Size	Lipids	Result/Conclusion
Yang et. al, 2022 [90]	Retrospective study	APIP patients (n = 190)	Triglyceride, total cholesterol	Total cholesterol is an independent predictor of moderately severe and severe AP in pregnancy.
Zhao et.al, 2022 [92]	Cohort study	antenatal mothers with AP (n = 45)	Triglycerides, total cholesterol	Total cholesterol levels were significantly higher in patients staying in hospital for >13 days compared with those staying for less than 13 days.
Shen et.al, 2021 [93]	Case-control study	Non-AP patients (n = 356) and AP patients (n = 349)	Total cholesterol, triglyceride, Apo A1	The occurrence of AP was significantly associated with total cholesterol.
D. Düzenci et.al, 2021 [94]	Retrospective study	AP patients (n = 153)	Total cholesterol, triglycerides, CRP	A significant correlation was observed between total cholesterol and mortality.
Song et.al, 2021 [95]	Retrospective study	non-recurrent AP patients (n = 671) and recurrent AP patient	Triglyceride, HDL-C, total cholesterol	The total cholesterol level was significantly higher in recurrent acute pancreatitis compared to AP at first onset.
Hong et.al, 2020 [14]	Retrospective study	AP patients (n = 648)	HDL-C, LDL-C, triglyceride	Both low total cholesterol (<160 mg/dL) and high total cholesterol (>240 mg/dL) levels within 24 h of admission are independently associated with an increased risk of SAP.
Peng et.al, 2015 [97]	Prospective study	patients predicted SAP (n = 66)	Total cholesterol, HDL-C, LDL-C, triglyceride	The total cholesterol level was significantly lower in AP patients with organ failure than without organ failure.
Khan et.al, 2013 [96]	Retrospective study	AP patients (n = 233)	Total cholesterol, HDL-C, LDL-C	Levels of serum total cholesterol are significantly lower in patients with SAP and are associated with longer hospitalization.
Wang et.al, 2010 [98]	Retrospective study	SAP patients (n = 338)	Total Cholesterol	Within 24 h after admission, the serum total cholesterol was a mortality-reduced factor when it is between 4.37 mmol/L and 5.23 mmol/L.

**Abbreviations:** Abbreviations: APIP, acute pancreatitis in pregnancy; AP, acute pancreatitis; HDL-C, high-density lipoprotein cholesterol; LDL-C, low-density lipoprotein cholesterol; SAP, severe acute pancreatitis; Apo A1, apolipoprotein A1; Apo B, apolipoprotein B; HTG, hypertriglyceridemia; CRP, C-reactive protein; RF, random forest; BISAP, Bedside Index of Severity in Acute Pancreatitis; OF, organ failure; ICU, intensive care unit.

**Table 2 jcm-12-01729-t002:** Characteristics of included clinical studies related to the relationship between LDL-C and AP.

Author and Year of Publication	Type of Study	Sample Size	Lipids	Result/Conclusion
Hong et.al, 2022 [91]	Retrospective study	Non-SAP patient (n = 438) and SAP patient (n = 49)	HDL-C, LDL-C	Based on a variable importance analysis of the RF model, LDL-C is an important predictor of SAP.
Liao et.al, 2021 [99]	Retrospective study	hypertriglyceridemic pancreatitis patients (n = 157)	Triglyceride, LDL-C	Higher LDL-C was associated with HTGP recurrence.
Wu et. al, 2019 [101]	Retrospective study	Non-SAP patient (n = 310) and SAP patient (n = 65)	Serum Apo B/A1 ratio	Apo B is the main structure of LDL-C. The serum Apo B/A1 ratio at admission is closely correlated with disease severity in patients with AP and can serve as a reliable indicator for SAP in the clinical setting.
Hong et. al, 2018 [13]	Retrospective study	AP patient (n = 647)	HDL-C, LDL-C	LDL-C (90 mg/dL) levels within 24 h of admission are independently associated with an increased risk of SAP.
Hong et.al, 2017 [100]	Case-control study	AP patients with amylase ≥300 U/L (n = 108) and AP patients with amylase <300 U/L (n = 108).	LDL-C, triglyceride	Higher LDL-C levels are independently associated with the development of AP along with low serum amylase.
Peng et.al, 2015 [97]	Prospective study	patients predicted SAP (n = 66)	Total cholesterol, HDL-C, LDL-C, triglyceride	The serum LDL-C level of patients with persistent organ failure was significantly lower than that of patients without persistent organ failure within 24 h of admission, and persistent organ failure is closely related to the high mortality rate of SAP.
Khan et. al, 2013 [96]	Retrospective study	Non-SAP patient (n = 30) and SAP patient (n = 203)	Serum total cholesterol, HDL-C, LDL-C	The level of LDL-C is significantly lower in patients with SAP and is associated with longer hospitalization.

**Abbreviations:** AP, acute pancreatitis; HDL-C, high-density lipoprotein cholesterol; LDL-C, low-density lipoprotein cholesterol; SAP, severe acute pancreatitis; Apo B, apolipoprotein B; Apo A1, apolipoprotein A1.

**Table 3 jcm-12-01729-t003:** Characteristics of included clinical studies related to relationship between HDL-C, Apo A1 and AP.

Author and Year of Publication	Type of Study	Sample Size	Lipids	Result/Conclusion
Hong et al., 2022 [91]	Retrospective study	Non-SAP patient (n = 438) and SAP patient (n = 49)	HDL-C, LDL-C	Based on a variable importance analysis of the RF model, HDL-C is the important predictor of SAP.
Li et al., 2021 [83]	Retrospective study	Non-SAP patient (n = 600) and SAP patient (n = 78)	Apo A1, HDL-C	Apo A1 and HDL-C levels were negatively correlated to the occurrence of SAP.
Zhou et al., 2018 [15]	Retrospective study	102 adult AP patients with OF, worsening of previous comorbidities or local complications during hospitalization	Apo A1, HDL-C	Apo A1 and HDL-C levels are negatively associated with some scoring systems and poor clinical outcomes in AP patients. The concentrations of Apo A1 and HDL-C have high predictive value to forecast persistent OF.
Hong et al., 2017 [62]	Retrospective study	Non-SAP patient (n = 589) and SAP patient (n = 58)	HDL-C	HDL-C at admission may predict development of SAP.
Peng et al., 2015 [97]	Retrospective study	66 patients with predicted SAP admitted to ICU	Apo A1, HDL-C	Low levels of Apo A1 and HDL-C are associated with high levels of inflammatory cytokines, persistent OF, infected necrosis and hospital mortality. Apo A1 and HDL-C can serve as biomarkers to differentiate persistent OF from transient OF.
Khan et al., 2013 [96]	Retrospective study	Non-SAP patient (n = 30) and SAP patient (n = 203)	Serum total cholesterol, HDL-C, LDL-C	The levels of HDL-C are significantly lower in patients with SAP and are associated with longer hospitalization.

**Abbreviations:** AP, acute pancreatitis; HDL-C, high-density lipoprotein cholesterol; LDL-C, low-density lipoprotein cholesterol; SAP, severe acute pancreatitis; RF, random forest; Apo A1, apolipoprotein A1; BISAP, Bedside Index of Severity in Acute Pancreatitis; OF, organ failure; ICU, intensive care unit.

## Data Availability

The datasets used or analyzed during the current study are available from the corresponding author upon request.

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
