# Peer review of "Relationship between Cholesterol-Related Lipids and Severe Acute Pancreatitis: From Bench to Bedside"

_jcm, 2023, doi:10.3390/jcm12051729_

Round 1
Reviewer 1 Report
This a very detailed review of an important and overlooked topic. Cholesterol metabolism may perpetuate the inflammation in acute pancreatitis leading to its severe form, which carries high mortality. This is a first review covering this topic. Authors made a huge effort in systematically leading us from what cholesterol is and how it is metabolized to how it is connected to severe acute pancreatitis and vice versa.
Since people reading Journal of clinical medicine are mostly clinicians, I think that `bench` part of the review is too long and out of the scope of this journal. It may be a matter of taste, but objectively, the preclinical language is vey difficult to read and follow and the reader is tired after so many pages (9 pages of `bench` vs. 3 pages of `bedside`). My suggestion is to dramatically shorten the chapter 1. to only most essential knowledge of basic cholesterol synthesis and metabolism (mostly written under Figure 1). Line 222 and 223 are repetitions and should be deleted. Paragraph from 242 till 259 should be shortened, in fact Figure 2 explanation is enough.
The `bedside` part is well written, very in depth. I would suggest some changes. In line 587 is not something that can be written just yet. More evidence is needed for hypercholesterolemia to be considered and etiology of acute pancreatitis, but it may be associated with more severe acute pancreatitis. Please change that. Also, I would introduce a table with all the clinical studies with associations between acute pancreatitis and cholesterol.
Author Response
Authors response: Thank you for your careful review of the manuscript and pointing out the shortcomings of the manuscript. I would like to appreciate your insightful comments and suggestions on the manuscript.
1.-My suggestion is to dramatically shorten the chapter 1. to only most essential knowledge of basic cholesterol synthesis and metabolism (mostly written under Figure 1)
Response: We have significantly shortened chapter 1, keeping the basic knowledge about cholesterol synthesis and metabolism. For instance, we deleted page 3 lines 124-131, page 5 lines 165-168, lines 176-178, lines 181-185, lines 187-191, lines 196-201 and page 6 lines 213-215.
2.-Line 222 and 223 are repetitions and should be deleted. Paragraph from 242 till 259 should be shortened, in fact Figure 2 explanation is enough.
Response: We deleted line 222 and 223.
We shortened paragraph from 242 till 259, now the new paragraph is “Cholesterol crystallization is one of the known stimuli of inflammasome activation…”.
3.-In line 587 is not something that can be written just yet. More evidence is needed for hypercholesterolemia to be considered and etiology of acute pancreatitis, but it may be associated with more severe acute pancreatitis.
Response: We have deleted the content of lines 587 and rephrased as:
“The current review indicated that hypercholesterolemia may be associated with AP 14.”.
4.-I would introduce a table with all the clinical studies with associations between acute pancreatitis and cholesterol.
Response: We have added three tables to show that.

Reviewer 2 Report
This rewiew mostly deals with cholesterol metabolism and its involvement in inflammation in general. However it deals much less with the putative relation between cholesterol and acute pancreatitis (AP). However it is a satisfactory compilation of data about cholesterol and inflammation. The English language of Chapter 4 should be improved.
Optional: The authors may add some information about cholesterol and procoagulant activity in AP taken into account the efficacy of LWMH in SAP.
Author Response
Authors response: Thank you for pointing out shortcomings in the manuscript. We have revised the manuscript according to your kind advices and detailed suggestions.
1.-The English language of Chapter 4 should be improved.
Response: We have polished the English expressions of chapter4.
2.-The authors may add some information about cholesterol and procoagulant activity in AP taken into account the efficacy of LWMH in SAP.
Response: We added this in chapter 2, page 7, as follows: “Pancreatic microvascular abnormalities and ischemia…”.
This section is also added in chapter 4, page 17:“It is well-known that coagulation abnormality…”.

Reviewer 3 Report
COMMENTS TO THE AUTHOR(S)
This is a review of ID: jcm-2172702, “Relationship between Cholesterol Related Lipids and Severe Acute Pancreatitis: From Bench to Bedside”. This is a very significant manuscript, but the overall volume is long and the main points are blurred
Major Comments:
This manuscript is very important because there are only a few studies on the etiology of pancreatitis limited to hypercholesterolemia. However, the biochemical information, e.g., HMG-CoA reductase, is reviewed. This is not linked to the main topic of the second half of the manuscript, which is acute pancreatitis and hypercholesterolemia. The volume of text is more conspicuous. How about omitting the basic contents and summarizing the basic experiments, animal experiments, and a few case reports? This is an excellent manuscript that addresses an extremely important topic in a sincere manner and should be made available to a much larger audience.
Minor comments
1) Page16, line585-587
There is no consensus that hyperlipidemia does not include hypertriglyceridemia. Consequently, there is no well-known definition of hyperlipidemia that is limited to the cause of pancreatitis and does not include hypertriglyceridemia.
2) Figure
The basic content is followed by a number of easy-to-understand figures. However, the current topic is the relationship between high cholesterol and acute pancreatitis, not cholesterol metabolism, which is not essential information.
It is helpful to summarise in the content of the figure what is currently evident and what is not evident about the relationship between high cholesterol and acute pancreatitis, including basic experiments, pathways in animal experiments and a small number of case reports.
Author Response
Authors response: Thank you for pointing out shortcomings in the manuscript. According with your advice, we amended the relevant part in manuscript. Some of your questions are answered below.
1.-The biochemical information, e.g., HMG-CoA reductase, is reviewed. This is not linked to the main topic of the second half of the manuscript, which is acute pancreatitis and hypercholesterolemia.
Response:We have partially removed biochemical information not linked to acute pancreatitis and hypercholesterolemia. For instance, we deleted page 3 lines 124-131, page 5 lines 165-168, lines 176-178, lines 181-185, lines 187-191, lines 196-201 and page 6 lines 213-215, lines 473-476.
2.-How about omitting the basic contents and summarizing the basic experiments, animal experiments, and a few case reports?
Response: We have made deletions and summarizing works in chapter 2 and chapter 3 for improvement.
Minor comments
1)Page16, line585-587
There is no consensus that hyperlipidemia does not include hypertriglyceridemia. Consequently, there is no well-known definition of hyperlipidemia that is limited to the cause of pancreatitis and does not include hypertriglyceridemia.
Response: We have deleted the content of lines 585-587 and rephrased the content: “Hypertriglyceridemia is the third most common cause of AP and have been widely studied. Triglyceride and cholesterol are both important participant in lipid homeostasis and likely intricately linked to the development and/or sustainment of inflammation. However, researches focusing on the hypercholesterolemia are still limited.”.
2)Figure
The basic content is followed by a number of easy-to-understand figures. However, the current topic is the relationship between high cholesterol and acute pancreatitis, not cholesterol metabolism, which is not essential information.
Response: We have shortened the content on cholesterol metabolism in chapter 2.
It is helpful to summarise in the content of the figure what is currently evident and what is not evident about the relationship between high cholesterol and acute pancreatitis, including basic experiments, pathways in animal experiments and a small number of case reports.
Response: We have added a new graphical abstract to demonstrate the present studies results, including basic experiments, pathways in animal experiments and a small number of case reports. The full lines represent the studies with much evidence, the dotted lines represent the studies with less evidence instead.

Round 2
Reviewer 1 Report
The manuscript is improved.